# Surveillance of Bacterial Load and Multidrug-Resistant Bacteria on Surfaces of Public Restrooms

**DOI:** 10.3390/ijerph21050574

**Published:** 2024-04-30

**Authors:** Khadega Ibrahim, Maisha Tahsin, Aninda Rahman, Shaikh Mizanoor Rahman, Md Mizanur Rahman

**Affiliations:** 1Flow Cytometry Core, Research Department, Sidra Medicine, Doha, Qatar; kibrahim2@sidra.org; 2Independent Researcher, Dhaka 1216, Bangladesh; maishatahsin31@gmail.com; 3Communicable Disease Control, Directorate General of Health Services, Ministry of Health & Family Welfare, Dhaka 1212, Bangladesh; dr.aninda@mis.dghs.gov.bd; 4Natural and Medical Sciences Research Center, University of Nizwa, Nizwa 616, Oman; shaikh.rahman@unizwa.edu.om; 5Biological Science Program, Department of Biological and Environmental Sciences, Collage of Arts and Sciences, Qatar University, Doha 2713, Qatar

**Keywords:** bacteria, public restrooms, Gram-positive/negative, antibiotic resistance, multidrug-resistant

## Abstract

Public restrooms are often a hub of microbial contamination and the examination of bacterial contamination in these facilities can serve as an important indicator of the transmission of infectious diseases. This study was conducted to determine the prevalence of bacterial contamination in public restrooms based on the economic class of the building. Samples were collected from various spots in 32 restrooms found in 10 shopping malls, classifying them into two categories: upper-end restrooms and lower-end restrooms. The findings showed that the level of contamination was higher in the lower-end restrooms, with the seat being the most contaminated area. The most dominant Gram-positive bacteria were of the coagulase-negative staphylococci species, making up 86% of the identified Gram-positive isolates. The most dominant Gram-negative bacteria identified were *Klebsiella pneumoniae *(*K. pneumoniae*)** and *Pseudomonas aeruginosa *(*P. aeruginosa*)**. The antibiotic sensitivity test results revealed the presence of multidrug-resistant bacteria among the Gram-positive and negative isolates, including *Staphylococcus haemolyticus *(*S. haemolyticus*)**, *Staphylococcus kloosii *(*S. kloosii*)**, *Acinetobacter baumanii *(*A. baumanii*)**, and *P. aeruginosa*. In conclusion, the study underscores the significance of monitoring bacterial contamination in public restrooms and the need for measures to reduce the spread of infectious diseases. Further research is crucial to gain a complete understanding of the bacterial contamination in public restrooms and their resistance patterns, to ensure the safety and health of the public. The implementation of improved cleaning practices and hands-free designs in addition to the installation of antimicrobial surfaces in restrooms can help reduce the risk of cross-contamination and prevent the spread of diseases.

## 1. Introduction

Fomites refer to non-living objects or substances capable of harboring infectious organisms, facilitating their transmission from one person to another. The degree of fomite contamination is influenced by factors such as moisture presence, frequency of usage, and hygiene or cleanliness. Fomites are notorious for being a major source of hospital-acquired infections and serving as a potential pathway for pathogens to spread between patients. Common fomites include door handles, showers, toilet seats and faucets, sinks, lockers, chairs, and tables. They are prevalent in various public spaces like hospitals, hotels, restaurants, and restrooms [1].

In general, the risk of infection spread via fomites depends on various factors: how often one comes into contact with the contaminated area, the number of microbes released by the infected individual, the likelihood of passing the infection to someone susceptible, the virulence and potency of the micro-organisms, the effectiveness of the immune system of those in contact, and the implementation of preventive measures such as using sanitizers/disinfectants and maintaining personal hygiene. As a result, people who rarely wash their hands after using the restroom may gain community-acquired methicillin-resistant *Staphylococcus aureus* (*S. aureus*) (CA-MRSA), which can cause an outbreak, particularly in places where they are highly prevalent. In addition, by aerosolization and direct transmission from hands to the fomite surface, fomites can act as a reservoir for bacteria and viruses [1].

The regular usage of public restrooms could have a substantial impact on the spread and transmission of infectious diseases and other bacterial contamination. Because many individuals use public restrooms or washbasins and touch doorknobs numerous times a day, contamination and pathogenic infectious diseases can spread. As a result, the significance of toilets and washbasins as a source of bacterial contamination and infections becomes clearer. It is self-evident that raising people’s awareness of transitory contamination and related diseases can benefit social health and prevent the spread of infections [2].

Many species of Gram-negative bacteria such as *Escherichia* and *Salmonella* and Gram-positive bacteria such as *Staphylococcus*, particularly MRSA and *Streptococcus*, can all be found in public restrooms [3]. They gain access to restrooms through human excrement (urine and faces) [4] or through human body contact, as many Gram-positive bacteria reside in many parts of the human body such as the skin, conjunctiva, nose, pharynx, mouth, lower gastrointestinal tract, anterior urethra, vagina, etc. [5]. Inadequate toilet cleanliness and improper toilet use might allow bacteria to spread from toilets to other areas. Toilet users’ contaminated hands can spread bacteria to the flushing handles, door handles, and faucets of toilets. The large amount of toilet flush aerosols produced while flushing can contaminate toilet seats and lids, adjacent floors, and neighboring surfaces. The pathogen’s capacity to live on many surfaces in the toilets provides a significant danger of infection to toilet users. The length of time a pathogen may survive on a surface varies depending on the pathogen. The majority of pathogens, such as the *Shigella* species, *Escherichia* species, *Clostridium* species, severe acute respiratory syndrome (SARS) coronavirus, and norovirus, can remain on the surfaces for weeks or even months [4]. 

Coliform bacteria are rod-shaped Gram-negative bacteria that do not have the ability to form spores and they are facultative anaerobes that ferment lactose quickly into acid and gas. In general, coliform bacteria include bacterial genera such as *Citrobacter freundii *(*C. freundii*)**, *Enterobacter cloacae *(*E. cloacae*)**, *Enterobacter aerogenes *(*E. aerogenes*)**, *Escherichia coli* (*E. coli*)**, and *Klebsiella pneumoniae* (*K. pneumoniae*), all of which belong to the Enterobacteriaceae family. Some coliform bacteria are called fecal coliforms and are found in the intestine (colon) of warm-blooded animals, whereas others are found in plants [4]. If coliform bacteria are found in food, this means that the conditions are favorable for the existence of enteric pathogens, and it may indicate that the sanitary precautions are insufficient. That is why coliform bacteria are used as a sanitation and hygiene indicator (fecal contamination indicator) micro-organism [6,7]. Normally, coliforms do not cause major sicknesses or diseases, but they can grow easily, and their presence can be a sign of the existence of other pathogenic organisms of fecal origin. Disease-causing bacteria, viruses, and protozoa, as well as many multicellular parasites, are examples of these pathogenic organisms [8].

Bacteria from public restrooms are a major problem to public health when they enter the body through hand-to-mouth or hand-to-food contact, causing illnesses [9]. Boils and food borne diseases caused by *S. aureus* and *E. coli* [4,10], urinary tract infections (UTIs) and diarrhea caused by *E. coli* and *Pseudomonas aeruginosa* (*P. aeruginosa*) [4,11], and sore throat caused by *Streptococcus pyogenes* (*S. pyogenes*) are among the bacterial diseases that can be transmitted through the use of restrooms [4]. 

If bacteria isolated from public restrooms demonstrate resistance to antibiotics, the issue will escalate, worsening the antibiotic resistance crisis. This is because the drug-resistant bacteria in this case are found in the publicly shared areas like public restroom surfaces which makes their transmittance easier. Antibiotic resistance is a worldwide public health concern as antibiotic-resistant bacteria are becoming increasingly common [12]. Antibiotic resistance makes some curable bacterial infections incurable as the ability to cure the bacterial infections in humans, or even animals and plants, will be decreased. This leads to more human illnesses, suffering, and even death, as well as increased treatment costs and duration, in addition to the increased side effects resulting from the usage of many and stronger medicines [12]. As few new antibiotics are being developed, antibiotics should be used carefully and only in the urgent cases [13].

The misuse and overuse of antibiotics leads to the development of antibiotic resistance by bacteria to be able to survive. The main and common antibiotic resistance mechanisms are the prevention of antibiotics accumulation through decreasing the uptake or increasing the efflux, alteration of the antibiotic target (i.e., ribosome subunits, cell wall penicillin binding proteins (PBPs), or DNA gyrase and topoisomerase IV), and antibiotic inactivation through enzymatic modification or degradation [14].

Regular hand washing and disinfectant cleaning of public restrooms at least two times a day are all suggested in the programs designed to control infections and decrease the risk caused by bacterial infections. Sensor-operated paper towel dispensers and touch-free electric hand dryers are two new technologies used to reduce the infections caused by the usage of public restrooms. The number of micro-organisms (bacteria) that are emitted into the air can be reduced by closing the toilet seat after usage [4].

Shopping malls are one of the most heavily frequented public spaces. The economic quality of these facilities varies as some are high-class luxurious malls which attract wealthy, high socioeconomic status individuals. Other malls are of a poorer quality, with poor maintenance and services and thus they are visited by individuals in lower socioeconomic groups or classes. As a result, due to its economic class, contamination in restrooms is predicted to be influenced by factors affecting the building quality, maintenance, and service excellence.

The aim of this study is to establish a baseline study about the degree and variety of bacterial contamination in public restrooms in shopping malls based on their economic status and healthcare quality. Particularly, the study aims to provide a qualitative and quantitative assessment about the prevalence of Gram-positive and Gram-negative bacteria in public restrooms using several tests, as well as their susceptibility and resistance patterns to a variety of antibiotic classes.

## 2. Materials and Methods

### 2.1. Sample Collection and Transportation

This cross-sectional observational study was conducted in Doha, Qatar. Dry sterile cotton swabs (Puritan Medical Products) were used to collect samples from public restrooms in 10 selected shopping malls, 5 lower-end and 5 upper-end, and from 5 spots, which are the seat (S), water sprayer (W), tap (T), inner door handle (ID), and outdoor handle (OD). They were opened in the restroom, dipped in the conical tube containing a peptone water medium (OXOID, Hampshire, UK), and rubbed across the surface (spot) of interest. Then, the swabs were returned to the conical tubes and the tubes were labeled according to the shopping mall number, restroom category (♂/♀), economic status (lower-end restroom (LR) or upper-end restroom (UR)), spot, and replicate number. Restrooms are categorized as UR when situated in malls with high-end brand shops and restaurants, and as LR when located in malls with local groceries and relatively cheaper restaurants. URs are mostly equipped with touch-less units and dedicated cleaning personnel are present to clean the restroom after each use, whereas in LRs, cleaning is typically less frequent, often occurring at the end of the day or the beginning of the next morning. The samples that were reported were collected during the weekend between 11 am and 1 pm, which corresponds to a moderately busy period, typically considered a medium rush hour. All tubes were placed on ice and taken to the lab for further plating and processing. For standardization, a weekend day was chosen for the sample collection. The standardization was conducted by observing the bacterial growth at rush and non-rush hours, and accordingly, a time in-between was chosen for the rest of the collections.

### 2.2. Dilution, Plating, and Incubation 

In less than 24 h, the samples were processed in the lab. Firstly, the tubes were vortexed for 40 s (if needed) to disrupt the precipitation at the bottom of the tubes. For the samples targeting Gram-positive bacteria, a serial dilution up to 10^−5^ dilution was performed for all samples for standardization purposes. For each sample, both the original sample as well as the 5 diluted samples were plated. Each sample was aseptically plated on two nutrient agar (NA) (Thermo Fisher Scientific, Cambridge, UK) plates. The plates were incubated upside down in a 37 °C incubator for 48 h. Regarding the samples targeting Gram-negative bacteria, the plating process was performed as mentioned before but the plates used were MacConkey agar (MAC) plates instead of the NA plates. 

### 2.3. Counting and Characterization

After 48 h, the plates were taken from the incubator to count the colonies and record their number in order to calculate the colony forming unit (CFU) for the samples targeting Gram-positive bacteria and the total coliform count for the samples targeting Gram-negative bacteria. For the samples targeting Gram-positive bacteria, the most dominant isolates (24 isolates) were sub-cultured on new NA plates. However, for the samples targeting Gram-negative bacteria, the colonies were not sub-cultured. The bacterial isolates were characterized morphologically by identifying the form or shape, surface, color, margins, and/or elevation.

### 2.4. Identification of the Isolates

For the samples targeting Gram-positive bacteria, the 24 most dominant types of colonies or isolates underwent Gram staining to select the Gram-positive isolates for further processing and identification. A series of identification methods were performed to identify the types of Gram-positive bacteria in the samples. Three identification methods were used, which are the primary identification methods by conventional and biochemical tests, matrix-assisted laser desorption ionization–time of flight mass spectrometry (MALDI-TOF-MS) rapid identification, and the confirmatory identification by the BD Phoenix™ automated microbiology system. The identification process for the samples targeting Gram-negative bacteria was performed using MALDI-TOF-MS after selecting 5 bacterial isolates and sub-culturing them on NA plates.

Conventional and Biochemical Tests: These tests include Gram staining, the catalase production test, carbohydrate (glucose/arabinose) fermentation test, and selective media (mannitol salt agar (MSA) and MAC) tests.

MALDI-TOF-MS Analysis: The most dominant isolates were purified on NA plates one day before and analyzed using MALDI-TOF-MS. The purified isolates were transferred to the MALDI target plate and a matrix solution, an energy absorbent organic substance, was added to the samples to be analyzed by MALDI-TOF-MS. The resulting spectrum for each sample was analyzed by the MALDI Biotyper (MBT) Compass Software (RUO/GP, Bruker Daltonik, Bremen, Germany) and a molecular fingerprint peptide mass fingerprint (PMF) was generated from the mass spectrum giving a species-specific pattern. This software assesses each spectrum compared to a reference spectrum in the database to determine the best match for each sample. A score (QI) between 0 and 3 was given to each sample to compare the level of similarity between the pattern given by the unknown sample and the database where the higher similarity is represented by a higher score (closer to 3). This test was repeated twice for confirmation.

BD Phoenix™ Identification Test: The samples tested using MALDI-TOF-MS were double tested by the BD Phoenix™ automated microbiology system as a confirmatory identification test.

### 2.5. Antimicrobial Susceptibility Testing (AST) 

Antibiotic susceptibility and resistance patterns of the isolates were studied using BD Phoenix^TM^ automated microbiology system. This system primarily determines phenotypic resistance patterns for antibiotic susceptibility and resistance. This system gives susceptible-, intermediate-, and resistant (SIR)-based interpretations. Selected samples from the samples targeting Gram-positive bacteria were tested by two resistance markers which are the phoenix methicillin-resistance in *Staphylococci* (MRS) and the beta lactamase producing bacteria (BLACT) markers. *Staphylococci* spp., which are frequently isolated from restrooms, were chosen to be tested against 23 different antibiotics. In the case of samples targeting Gram-negative bacteria, 21 antibiotics were used.

### 2.6. Statistical Analysis

The statistically significant differences were determined using the Student’s unpaired *t* test using a 95% confidence level (significance level (α) = 0.05). The software used to analyze the data was GraphPad Prism 9. In all cases, a *p*-value less than 0.05 (* *p* ˂ 0.05) was considered significant, and the values were expressed as “mean ± SD”.

## 3. Results

### 3.1. Contamination Level and Diversity Assessment

The contamination level of the restrooms was represented as CFU values for the samples targeting Gram-positive bacteria and as the total coliform count for the samples targeting Gram-negative bacteria. The comparison of contamination levels showed slight variations under the same category of restroom economic class. However, in general, the contamination level in the LR category was significantly higher in all spots compared to the UR category, with the highest contamination level shown in the “S” spot (Figure 1). Moreover, for the sample targeting Gram-positive bacteria, the bacterial diversity in the LR category was higher than in the UR as the number of the different types of isolates/colonies found in the LR’s randomly selected “T” spot sample was 14, which is double the number of isolates found in the UR’s random sample. Regarding the samples targeting Gram-negative bacteria, the LR category had a higher diversity with 34 different types of colonies according to the morphological characterization of all observed isolates. However, the UR category had only 11 different isolates. Overall, the LR category had a higher contamination level as well as higher bacterial diversity than the UR category.

### 3.2. Conventional and Biochemical Test Identification

The results of Gram staining showed that Gram-positive bacteria represented 18 isolates out of the 24 most dominant isolates, which means that 75% of the isolates were Gram-positive bacteria. The summarized results for the biochemical tests for the 18 Gram-positive isolates are shown in Table 1. The results of the performed tests suggested the presence of *S. aureus* in many restrooms in addition to the presence of *Bacillus* and *Micrococcus* species.

### 3.3. MALDI-TOF-MS Identification

The results of the MALDI-TOF identification test for the six selected Gram-positive isolates among the 18 isolates (Table 1) are shown in Table 2. The six expected bacterial isolates were *Staphylococcus haemolyticus* (*S. haemolyticus*), *Staphylococcus kloosii* (*S. kloosii*), *Micrococcus leuteus* (*M. leuteus*), *Staphylococcus pasteuri* (*S. pasteuri*), *Bacillus clausii* (*B. clausii*), and *Streptomyces violaceoruber* (*S. violaceoruber*). The most dominant isolate was *S. haemolyticus* with a percentage of 75% isolated from all malls, followed by *S. kloosii* with a percentage of 10% and *M. leuteus* with a percentage of 9%. This indicates that the coagulase-negative *Staphylococcus* (CNS) species dominates the identified isolates with a percentage of 86% overall. The results of the MALDI-TOF identification test for the five dominant Gram-negative isolates and their prevalence are shown in Table 3. The five bacterial isolates identified were *K. pneumoniae*, *P. aeruginosa*, *Pantoea agglomerans* (*P. agglomerans*), *Acinetobacter baumanii* (*A. baumanii*), and *Acinetobacter lwoffii/haemolyticus* (*A. lwoffii/haemolyticus*). The most dominant isolate was *K. pneumoniae* with a percentage of approximately 87%, followed by *P. aeruginosa* with a percentage of approximately 9%.

### 3.4. BD Phoenix™ Identification Test

The same six selected Gram-positive isolates were further identified by the BD phoenix™ automated microbiology system and the results are shown in Table 4.

### 3.5. AST by BD Phoenix™

Resistance to antibiotics was detected both in Gram-positive and Gram-negative isolates. The antibiotic sensitivity pattern of the CNS species, which are the *S. haemolyticus* (from UR and LR), *S. kloosii*, and *S. pasteuri*, against the 23 antibiotics is shown in Table 5. Two out of three Gram-positive species were resistant to antibiotics. The results indicate that *S. haemolyticus* (UR) and *S. kloosii* are multidrug-resistant (MDR) bacteria. *S. haemolyticus* (UR) was found to be resistant to six antibiotics which are Amoxicillin-Clavulanate, Ampicillin, Cefotaximc, Imipenem, Oxacillin, and Penicillin G. Surprisingly, *S. haemolyticus* isolated from the URs exhibits more resistance than the same species isolated from LRs, which showed resistance to Erythromycin only. *S. kloosii* was found to be resistant to the same antibiotics as *S. haemolyticus* (UR) in addition to Erythromycin. *S. pasteuri*, however, showed no resistance to any of the antibiotics used.

The antibiotic sensitivity pattern for the *K. pneumoniae*, *P. aeruginosa*, *P. agglomerans*, and *A. baumanii* against the 21 antibiotics is shown in Table 6. The results indicate that *P. aeruginosa* and *A. baumanii* are MDR bacteria. *P. aeruginosa* was found to be resistant to 10 antibiotics which are Amoxicillin-Clavulanate, Ampicillin, Cefoxitin, Ceftriaxone, Cefuroxime, Cephalothin, Ertapenem, Nitrofurantoin, Tigecycline, and Trimethoprim-Sulfamethoxazole. *A. baumanii* was found to be resistant to eight antibiotics which are Amoxicillin-Clavulanate, Ampicillin, Aztreonam, Cefoxitin, Cefuroxime, Cephalothin, Ertapenem, and Nitrofurantoin. However, *K. pneumoniae* and *P. agglomerans* showed resistance to one antibiotic only which was Ampicillin and Cephalothin, respectively.

## 4. Discussion

Identification of selected isolates using matrix-assisted laser desorption ionization–time of flight mass spectrometry (MALDI-TOF-MS) and other techniques revealed that the coagulase-negative *Staphylococci* (CNS) species including *S. haemolyticus*, *S. kloosii*, and *S. pasteuri* represented 86% of the identified Gram-positive isolates, with *S. haemolyticus* being the most dominant strain among them (75%). *M. leuteus* and *B. clausii*, represented 9% and 1% of the identified Gram-positive isolates, respectively. The most dominant identified Gram-negative bacteria were *K. pneumoniae* (87.35%), *P. aeruginosa* (8.74%), *P. agglomerants* (0.26%), *A. baumanii* (2.15%), and *A. lwoffii/haemolyticus* (1.50%). The frequent occurrence and dominance of *K pneumoniae* could be due to its ability to survive in various environmental conditions, including moist areas like toilets. This resilience might have allowed it to persist in public restroom settings. Qatar’s weather is typically hot and humid which might have supported this micro-organism to survive longer in the environment and increase the potential risk of infecting others. In addition, many individuals can carry *K. pneumoniae* asymptomatically. When these carriers use public toilets, they can unconsciously introduce the bacteria into the environment. 

The results of the antibiotic sensitivity test (AST) using the BD Phoenix™ automated microbiology system revealed that there are some MDR bacteria among the Gram-positive and Gram-negative isolates which are *S. haemolyticus*, *S. kloosii*, *P. aeruginosa*, and *A. baumanii*. Although some of the isolated bacteria are members of the human microbiota, they can still be pathogenic or opportunistic pathogens, leading to numerous infectious diseases. Consequently, their resistance pattern as MDR bacteria can affect the future of treatment and transmission of infectious diseases.

Due to their warm and humid atmosphere, public restrooms are shown to have the ideal conditions for the accumulation of pathogenic or non-pathogenic micro-organisms, especially bacteria. In general, the LR category of the restrooms contains higher contamination levels compared to the UR category. Some factors which can influence the contamination degree in public restrooms are the frequency of cleaning shifts, especially during rush hours, and the quality of cleaning products. In contrast to the LR category, the URs are cleaned after every single use. Moreover, the building design can have a huge effect and it differs between the UR and LR categories. Unlike the LR category, it was noticed that most restrooms in high-class shopping malls do not have outer doors; instead, they have halls leading to the restroom area, and the sink taps turn on automatically by laser sensors, instead of manual tap handles. Consequently, the designs of the UR category lowered the number of touched surfaces in the restrooms, thus less bacterial contamination was observed. In general, these observations about restroom design differences regarding the “T” and “OD” spots might explain the reason why the contamination level in these spots was low especially in the UR category as these spots/surfaces are not touched frequently. This agrees with a study conducted by the National Research Council that links the spread of infections to the features of buildings. The study has addressed a number of factors that contribute to the contamination transmission in school buildings and their restrooms, and these factors include surface sanitizing, and the number and availability of touched surfaces like sinks and toilets. They suggested the replacement of traditional designs of door handles, flushers, and soap and towel dispensers with alternative hands-free designs in order to eliminate contamination spread, which might help the infectious disease transmission [15]. Furthermore, the “S” spot, especially in the LR category, was found to be hugely contaminated with both Gram-positive and Gram-negative bacteria as indicated by the uncountable growth, and this is possibly because this spot has a high contact frequency by restroom users. 

An observational study looked at the sanitation of public restrooms and tested the facilities’ (i.e., handwashing and hand-drying facilities) microbes. This study found that high- and/or middle-category restrooms were significantly more likely than low-category restrooms to have toilet seat disinfectants as well as a cleaner environment in the toilet/urinal area, floor areas, walls, and sinks [9]. Another study examined the bacterial contamination of regularly handled surfaces, such as toilet surfaces, in four shopping malls in the United Arab Emirates (UAE). According to this study, a lot of people of diverse ages, cultures, social classes, and, of course, different hygiene habits use the public restrooms in the UAE shopping malls, making them significantly more likely to be contaminated with bacteria than restrooms elsewhere. Additionally, the results show that mall cleaning and sanitization procedures must be changed to accommodate visitor density, with weekends and holidays necessitating more frequent cleaning effort than during regular weekdays [16].

The diversity of the bacterial taxa present in toilets, which is influenced by the number of different human occupants each day, can generally have an impact on the bacterial pollutants [17]. Due to physiological differences in the normal microbiota, gender can have an impact on diversity too.

The qualitative analysis of bacterial isolates from the numerous areas swabbed in the current experiment showed the abundance of the typical skin flora. Non-pathogenic bacteria of the skin flora, such as the Coagulase-negative *Staphylococcus* (CNS) species including *S. haemolyticus*, *S. kloosii*, and *S. pasteuri* were identified. The CNS was the most dominant type of species among the identified isolates [18]. This finding is consistent with research conducted at the Michael Okpara University of Agriculture, Umudike, to isolate, identify, and evaluate the pattern of antibiotic sensitivity of bacterial contaminants recovered from door handles, including of restrooms. According to this study, CNS species make up 21.2% of the 130 bacterial isolates in total [19]. In addition, the UAE study previously cited indicated that 99% of all positive samples included non-pathogenic skin bacteria such as *Staphylococcus epidermidis* (*S. epidermidis*) and other CNS species [16]. However, antibiotic resistance in non-pathogenic micro-organisms can still play a significant role by serving as a reservoir of resistance genes and can transfer these resistant genes horizontally to pathogenic bacteria, contributing to the spread of antibiotic resistance in clinical settings. They may contribute to the contamination of soil, water, or food with antibiotic-resistant genes, posing risks to human health and agricultural practices. 

Given that the majority of the surfaces studied come into close touch with human skin, the presence and dominance of the skin microbiota on toilet surfaces is not unexpected and is predicted. Research has shown that skin-associated bacteria are usually resilient and have the capacity to survive and remain on surfaces for extended periods of time [20]. Despite having a reputation as skin commensals, CNS species are the most common endemic nosocomial pathogen in newborns. Bloodstream infections, which cause 51% to 78% of newborns with very low birth weights (VLBW), make up the majority of CNS infections. CNS pathogens, however, have a low fatality rate and low pathogenicity [21]. *S. aureus* was suspected to be among the isolates, as shown by the biochemical tests, because of its traits as a Gram-positive, cocci-shaped, catalase-positive, glucose, mannitol, and lactose fermenter as well as its inability to grow in a MAC medium [18,22]. 

At the Sokoine University of Agriculture in Morogoro, Tanzania, a study was conducted to isolate, identify, and ascertain the bacterial loads in the public restrooms of the student residences. This study demonstrated the presence of MRSA, a drug-resistant form of *S. aureus*, as well as other bacteria, in public restrooms. Through human waste (faces and pee), they gain entry to the restrooms. Bacteria may spread from the toilets to adjacent locations due to poor toilet hygiene and incorrect toilet use. The flushing handles, door handles, and faucets of toilets can become contaminated with bacteria from the hands of toilet users [4]. Because *S. aureus* is thought to be a component of human skin and mucosal membranes and because people are their main reservoir, its presence in public restrooms is anticipated. It can spread from person to person through direct touch or fomites. As one of the most common bacterial infections in humans, it is also the root cause of a number of illnesses, including bacteremia, infective endocarditis, skin and soft tissue infections, osteomyelitis, septic arthritis, infections of prosthetic devices, pulmonary infections, etc. [23]. Additionally, certain locations in the restrooms had *Bacillus* spp. Because they are spore-forming bacteria, these species can endure extreme temperatures, cold, radiation, desiccation, and chemical disinfectants [19].

Some of the detected isolates fall into the taxa/phyla linked with the gut. For instance, the *Staphylococci* species are members of the Firmicutes phylum, which together with the Bacteroidetes phylum, accounts for 99% of the gut microbiota [24]. Taxa associated with the gut were more prevalent on toilet surfaces, indicating that feces had been present there. Indirect contact with water splashes (aerosols) from toilet flushing can also result in fecal contamination, as can direct contact with feces or dirty hands. Given that enteropathogenic bacteria may spread similarly to how human commensals do, the high number of gut-associated species found in restrooms is alarming for the general public’s health [25].

Other studies focusing on non-hospital environments with equally diverse bacterial populations have mentioned that restroom environments are a possible harbor for bacterial populations with an antibiotic resistance and suggested that the diverse bacterial populations can provide the favorable conditions and environments to support the development, sustainability, and spread of bacterial antibiotic resistance. In addition to the prevalence of human diseases in restrooms, these studies have also suggested that restroom environments are a possible harbor for bacterial populations with an antibiotic resistance. Additionally, cells might be able to resist and survive under such conditions even if resources are scarce. Overall, this study indicated that toilets are one of the primary and potential sources of infections and that they may be able to sustain bacterial “resistomes” [3].

CNS species are clinically significant, firstly due to their nosocomial pathogenicity and secondly because many strains of the CNS species are methicillin-resistant and are developing more antibiotic resistance, which make CNS a serious problem [26], especially their resistance to consistently and commonly used antibiotics [27]. The findings of this investigation support the notion that *S. haemolyticus* is a MDR bacterium because it displayed resistance to seven different drugs. In the current investigation, *S. kloosii* also demonstrated resistance to seven different antibiotics. This finding partially accords with a study that examined the antibiotic resistance profile of *S. kloosii* isolated from a blood culture of a patient with sepsis and an intracranial hemorrhage using the disc diffusion method. Penicillin, oxacillin, erythromycin, clindamycin, cotrimoxazole, ofloxacin, and linezolid were all ineffective against *S. kloosii* [28]. The *S. pasteuri* isolate used in this investigation did not exhibit resistance to any of the examined antibiotics. A different study, however, found that *S. pasteuri* was resistant to a variety of antibiotics, including Chloramphenicol, Streptomycin, Fosfomycin, Macrolides, Lincosamides, Streptogramins, and Tetracyclines [29].

Regarding Gram-negative bacteria, the study conducted in Tanzania revealed that *P. aeruginosa* was more common (13.3%) than *K. pneumonia* (11.6%) in public bathrooms. These results are consistent with and comparable to those of this study. The isolates discovered in this study can also spread or cause a variety of illnesses. For instance, *K. pneumonia* produces pneumonia while *P. aeruginosa* causes UTI [4]. A Gram-negative bacterium that is connected with plants called *P. agglomerans* has the potential to infect people through open wounds and, in severe circumstances, can lead to septic arthritis. It is not always an infectious agent in people. It could, however, be a source of opportunistic human infections, mainly in immunocompromised people, through wound infection with plant material or as a hospital-acquired infection [30]. *P. agglomerans* was the least prevalent isolate in this study (0.26%) which is most likely due to it being a plant-associated bacteria. *A. baumanii* is a Gram-negative bacterium that can cause bacteremia, a disease when germs are present in the patient’s circulation, as well as a number of infections affecting the urinary, gastrointestinal, and respiratory tracts. This isolate displayed Penicillin and Cephalosporin resistance characteristics [31]. We found a similar antibiotic resistance pattern with this isolate in this study. Similarly, *A. lowffii/haemolyticus* are also Gram-negative isolates that can cause bacteremia [32]. Globally, *K. pneumoniae* is a common cause of MDR infections. Recent research has revealed that there is an MDR *K. pneumoniae* strain that is resistant to the last-line antibiotic colistin [33]. Furthermore, carbapenem-resistant *K. pneumoniae* (CRKP), which is resistant to carbapenems, a class of commonly used broad-spectrum antibiotics (such as Imipenem, Meropenem, and Ertapenem in our current study), poses a substantial challenge and is a growing concern in healthcare settings [34]. However, the results of this study revealed that the isolated strain of *K. pneumoniae* is neither multi-drug resistant (MDR) nor carbapenem-resistant; it is only resistant to ampicillin. 

In recent years, there has been increasing concern about the ability of MDR bacteria to evade the killing effect of disinfectants, which are commonly used to control the spread of infectious pathogens. One of the most notable discoveries in this study is the identification of highly MDR bacteria in an upscale restroom where a designated individual regularly disinfects the toilet after each use. This highlights the ability of MDR bacteria to withstand the disinfectant’s killing effect. Despite the significantly lower bacterial count in high-end restrooms compared to low-end ones, they are not entirely free from MDR bacteria. This underscores the importance of public awareness and the necessity of maintaining good hygiene practices when using public restrooms. Continued research and vigilance are essential to address this growing threat and develop effective strategies to combat disinfectant-resistant MDR bacteria. The increase in bacterial resistance resulting from the overuse of antibiotics underscores the need to investigate innovative antimicrobial strategies. One promising approach to address this growing challenge of antibiotic resistance is the development of new antibiotic classes that target bacterial metallophores, which are crucial for bacteria to acquire the metals necessary for their growth and survival [35].

The current study encountered some limitations. Due to the enormous number of bacterial colonies that were recovered from numerous locations, not every isolated colony was identified. In contrast, a small number of colonies were picked for identification based on their frequency and shape. It is possible that the bacterial species found in the current study do not accurately represent the species distribution in the sites examined. Additionally, due to a lack of resources, antibiotic sensitivity patterns were only performed on a small number of chosen isolates. Additionally, the layout of the facilities as well as the placement of the paper towel and hand soap dispensers were not monitored. This is crucial because, despite the fact that hand soaps, detergents, and paper towels are provided for bathroom users, they may occasionally be hidden from view or positioned incorrectly. Lately, there has been a keen focus on incorporating antimicrobial surfaces into the design of public restrooms. These surfaces, whether in the form of specialized materials or coatings, are specifically crafted to deter the growth and transmission of micro-organisms like bacteria, viruses, and fungi on commonly used restroom surfaces. They are engineered to provide an extra layer of defense against microbial contamination, thereby elevating hygiene standards. Modern public restrooms may integrate antimicrobial features such as coatings infused with silver or copper ions, antimicrobial paints, tiles, plastics, sealants, and fabrics, all aimed at enhancing restroom hygiene.

## 5. Conclusions

The results of this study raise questions about users’ awareness of cleanliness in shared public restrooms. The current study has demonstrated the amount and variety of bacterial contamination in public restrooms found in shopping malls and categorized based on their economic class into LR and UR. A very high level of Gram-positive and Gram-negative bacterial contamination is present in public restrooms on different spots. Although a small number of isolates were tested, a number of drug-resistant bacteria were isolated, which is really alarming. The future of illness treatment and transmission may be impacted by these contaminated areas acting as reservoirs for bacteria that are resistant to antibiotics. The isolates represent human microbiota and can be easily transferred among different individuals by making contact with contaminated surfaces. Some of the isolates were discovered to be MDR isolates and may be pathogenic. For a better quality of life, and better public health, individuals need to limit their contact with contaminated surfaces at public restrooms and it is highly recommended that restrooms should be designed to minimize the number of touchable surfaces that can spread contamination between restroom users, and that these should be replaced with hands-free (automated) designs. Additionally, the incorporation of antimicrobial surfaces in public restrooms could significantly contribute to minimizing the transmission of harmful bacteria, fungi, and viruses. These surfaces, infused with antimicrobial agents, are essential for proactive infection control measures. They effectively impede the proliferation and persistence of micro-organisms, thereby enhancing hygiene standards and fostering a healthier restroom environment for users. Moreover, raising public awareness about good hygiene practices is crucial in complementing these efforts.

## Figures and Tables

**Figure 1 ijerph-21-00574-f001:**
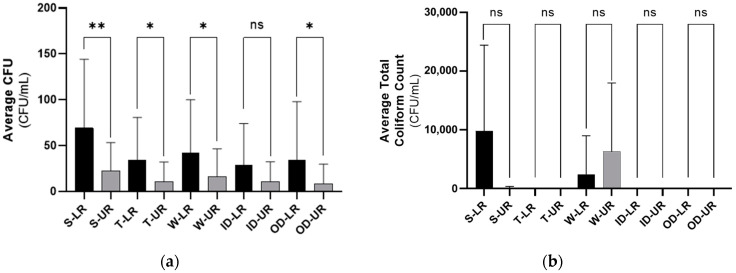
The difference in the average (**a**) CFU and (**b**) total coliform count between LR and UR categories for each spot. CFU and total coliform count values represent the contamination level in the restrooms, and they are expressed as “mean ± SD”. The statistical analysis was conducted using the Student’s unpaired *t* test. *p* < 0.05 is considered significant; ns, not significant; *, *p* ˂ 0.05; and **, *p* ˂ 0.01. CFU: colony forming unit, LR: lower-end restroom, UR: upper-end restroom, S: seat, W: water sprayer, T: tap, ID: inner door handle, OD: outer door handle. The total coliform count could not be calculated accurately due to the uncountable overgrowth on some “S” spot culture plates.

**Table 1 ijerph-21-00574-t001:** Summarized results for the biochemical tests of the 18 Gram-positive isolates.

Isolate Code	Total No. of Colonies	(%)	Gram Staining (+/−)	Microscopic Morphology	Catalase Test (+/−)	Glucose Test		Arabinose Test		MSA Test		MAC Test	
						GP	F	GP	F	GP	F	GP	F
1W	10,463	74.918	+	Clusters Cocci	+++	−	++	−	+	+++	++	−	−
2C	1383	9.903	+	Paired Cocci	++	−	++	−	++	−	−	+	brown
3Y	1221	8.743	+	Clusters Cocci	+++	−	+	−	+	++	−	−	−
8O	143	1.024	+	Clusters Cocci	+++	−	++++	−	+	++	++	−	−
17DO	100	0.716	+	Streptobacillus	+	−	++	−	+	−	−	−	−
16WBCI	71	0.508	+	Sporulating Bacillus	+	−	++	−	+	++	++	−	−
12BO	56	0.401	+	Streptococci	+	−	++	−	++	−	−	−	−
5WSF	23	0.165	+	Cocci	+++	−	+	−	+++	+++	++	++	++
19FF	23	0.165	+	Bacillus	+	−	++	−	++++	++	++	−	−
9TO	14	0.100	+	Oval	++	−	+	−	+	−	−	−	−
6WBF	13	0.093	+	Rod	++	−	+	−	+/−	+++	+	−	−
10RY	13	0.093	+	Clusters Cocci	+++	−	+	−	+	++	−	−	−
5RW	6	0.043	+	Sporulating Bacillus	++	−	+	−	+	−	−	++	++
11T	3	0.021	+	Streptobacillus	+++	−	+++	−	+	++	−	−	−
YBB	2	0.014	+	Cocci	++++	−	+	−	+	−	−	−	−
7R	1	0.007	+	Cocci	+++	−	+	−	+	+	−	−	−
13FW	1	0.007	+	Bacillus	++	−	+	−	+	++	++	−	−

GP: gas production, F: fermentation (color change), Gr: growth (tolerance), MSA: mannitol salt agar, MAC: MacConkey. −: no change (negative result), +: slight change, ++: moderate change, +++: high change, and ++++: very high change.

**Table 2 ijerph-21-00574-t002:** Identification of the six selected Gram-positive bacterial isolates using MALDI-TOF-MS.

Isolate Code	Expected Bacterial Strain	* Unknown Isolate MALDI-TOF Score
1W	*S. haemolyticus*	2.18 (+++)
2C	*S. kloosii*	No Matching
3Y	*M. leuteus*	1.97 (+)
8O	*S. pasteuri*	1.85 (+)
11T	*B. clausii*	1.92 (+)
7R	*S. violaceoruber*	1.86 (+)

* A score between 0.000 and 3.000 was given for each sample to compare the level of similarity between the pattern given by the unknown sample and the database, where the higher similarity is represented by a higher score (closer to 3.000). +: low similarity, +++: high similarity, and No Matching: no similarity with a known reference in the MALDI-TOF database.

**Table 3 ijerph-21-00574-t003:** Identification of the five most dominant Gram-negative bacterial isolates using MALDI-TOF-MS.

Isolate Code	Total Number of Colonies in All Malls	Prevalence Percentage (%)	Unknown Isolate Best Match
M4: ♀ W2	26,780	87.35	*K. pneumoniae*
M1: ♀ T1	2680	8.74	*P. aeruginosa*
M3: ♀ T2	80	0.26	*P. agglomerans*
M2: ♀ S1S2A	660	2.15	*A. baumanii*
M3: ♀ S2A	460	1.50	*A. lwoffii/haemolyticus*

**Table 4 ijerph-21-00574-t004:** BD Phoenix™ confirmation identification for the six selected Gram-positive bacterial isolates.

Isolate Code	PHOENIX ID
1W	*S. haemolyticus*
2C	N/A
3Y	*M. leuteus*
8O	*S. pasteuri*
11T	N/A
7R	*S. violaceoruber*

N/A: not available.

**Table 5 ijerph-21-00574-t005:** Antibiogram of four Gram-positive isolates against 23 antibiotics.

	Isolate	*S.s haemolyticus* (UR)		*S. haemolyticus* (LR)		*S. kloosii* (UR)		*S. pasteuri* (UR)	
Antibiotic		MIC	SIR	MIC	SIR	MIC	SIR	MIC	SIR
Amoxicillin-Clavulanate		≤1/0.5	R	≤1/0.5	S	≤1/0.5	R	≤1/0.5	S
Ampicillin			R				R		
Cefotaximc		16	R	≤8	S	≤8	R	≤8	S
Cefoxitin		>8		≤2		≤2		≤2	
Ciprofloxacin		≤0.5	S	≤0.5	S	≤0.5	S	≤0.5	S
Clindamycin		≤0.5	S	≤0.5	S	≤0.5	S	≤0.5	S
Daptomycin		≤1	S	≤1	S	≤1	S	≤1	S
Erythromycin		≤0.25	S	>4	R	>4	R	≤0.25	S
Fusidic Acid		>8		≤1		4		≤1	
Gentamicin		≤2	S	≤2	S	≤2	S	≤2	S
Gentamicin-Syn		≤500		≤500		≤500		≤500	
Imipenem		≤2	R	≤2	S	≤2	R	≤2	S
Linezolid		≤1	S	≤1	S	≤1	S	≤1	S
Moxifloxacin		≤0.5	S	≤0.5	S	≤0.5	S	≤0.5	S
Mupirocin-High level		≤256	S	≤256	S	≤256	S	≤256	S
Nitrofurantoin		≤16	S	≤16	S	≤16	S	≤16	S
Oxacillin		2	R	≤0.25	S	0.5	R	≤0.25	S
Penicillin G			R				R		
Rifampin		≤0.5	S	≤0.5	S	≤0.5	S	≤0.5	S
Teicoplanin		4	S	≤1	S	4	S	≤1	S
Tetracycline		≤0.5	S	≤0.5	S	≤0.5	S	≤0.5	S
Trimethoprim-Sulfamethoxazole		≤1/19	S	≤1/19	S	≤1/19	S	≤1/19	S
Vancomycin		1	S	≤0.5	S	1	S	1	S

UR: upper-end restroom, LR: lower-end restroom, MIC: minimal inhibitory concentration, SIR: susceptible, intermediate, resistant, R: resistant, S: susceptible.

**Table 6 ijerph-21-00574-t006:** Antibiogram of four Gram-negative isolates against 21 antibiotics.

	Isolate	*K. pneumoniae* (UR)	*P. aeruginosa* (UR)	*P. agglomerans* (LR)	*A. baumanii* (LR)
Antibiotic		SIR			
Amikacin		S	S	S	S
Amoxicillin-Clavulanate		S	R	S	R
Ampicillin		R	R	S	R
Aztreonam		S	S	S	R
Cefepime		S	S	S	S
Cefoxitin		S	R	S	R
Ceftazidime		S	S	S	S
Ceftriaxone		S	R	S	S
Cefuroxime		S	R	I	R
Cephalothin		S	R	R	R
Ciprofloxacin		S	S	S	S
Colistin		S	S	S	S
Ertapenem		S	R	S	R
Gentamicin		S	S	S	S
Imipenem		S	S	S	S
Levofloxacin		S	S	S	S
Meropenem		S	S	S	S
Nitrofurantoin		S	R	S	R
Piperacillin-Tazobactam		S	S	S	S
Tigecycline		S	R	S	S
Trimethoprim-Sulfamethoxazole		S	R	S	S

UR: upper-end restroom, LR: lower-end restroom, SIR: susceptible, intermediate, resistant, R: resistant, S: susceptible.

## Data Availability

Data is contained within the article.

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
