# Peer review of "Surveillance of Bacterial Load and Multidrug-Resistant Bacteria on Surfaces of Public Restrooms"

_ijerph, 2024, doi:10.3390/ijerph21050574_

Round 1

Reviewer 1 Report

Comments and Suggestions for Authors

The study by Khadega et al. is an interesting series of investigations into bacterial contamination on surfaces in public restrooms in shopping centers. Even if no surprising new findings are obtained, such a study is always good to draw attention to hygiene measures. Test material was taken from 32 toilets in 10 shopping malls, divided into upper-end and lower-end toilets. Among the Gram-positive species, the genus Staphylococcus predominated as a typical skin bacterium. K. pneumoniae was the most frequently isolated Gram-negative species. The bacteria were isolated after cultivation on culture media and the species affiliation was determined by biotyping. Antibiotic resistance was also analyzed.

As described, there are numerous parameters that result in a higher or lower bacterial load on surfaces. The frequency of use of these restrooms plays a major role here. Is there information on how many people have visited the restrooms after cleaning and up to the time of sampling? There may be many more people out at the weekend than during the week. And, how often were the restrooms be cleaned within a day?

Is the identified antibiotic resistance due to a genetic background, i.e. do the isolates carry these genes or is it a phenotypic resistance, which is very common in P. aeruginosa. A few more details should be added and specified here.

The frequent occurrence and dominance of Klebsiella pneumoniae is interesting. Is there an explanation for this or could this frequent occurrence also be shown in other studies?

Table 2: requires some explanation: what does "no matching" mean for S. kloosi? What do “+++” and “+” mean and which classifications were chosen?

Table 2: Change MULDI to MALDI

The score values are relatively low in order to be able to indicate a clear species affiliation. The MALDI-TOF method also determines other closest affiliations. Were the first-named species always species-identical or were other relatives also indicated? The procedure for determining the exact species affiliation of environmental microorganisms should be considered carefully.

In Tables 5 and 6, the bacterial names in the heading can be deleted as they are mentioned in the table. The species name of the bacteria should be written in lower case (Klebsiella pneumonia, not Pneumoniae)

The study shows that non-pathogenic germs have been isolated, particularly in the case of staphylococci. However, antibiotic-resistant bacteria were also found. This raises the question of a risk assessment of a possible transmission of the bacteria to other people associated with a risk of infection. Are there any figures or statistics that show how many people have become infected or contaminated in toilets and are therefore at an increased risk of transmission? This should clarify whether there is a high risk of infection and therefore an urgent need for action or whether simple hygiene measures should be taken to greatly reduce the risk of infection.

In what form does antibiotic resistance of a non-pathogenic microorganism play a role?

The different results of the upper-end and lower-end toilets are interesting. It shows that the touchless units are more hygienic. In addition to this requirement, an appeal should also be made to visitors with regard to good hygiene, as described. Another point that could certainly be included in the discussion is the installation of antimicrobial surfaces.

Minor comments:

Line 77: Citrobacter and the others count as coliforms and not the other way around.

Line 151: the nutrient agar should be specified.

Lines 159/168: isn't this just a repetition?

Line 320: not all bacteria mentioned are pathogenic, some are opportunistic pathogen.

Author Response

The study by Khadega et al. is an interesting series of investigations into bacterial contamination on surfaces in public restrooms in shopping centers. Even if no surprising new findings are obtained, such a study is always good to draw attention to hygiene measures. Test material was taken from 32 toilets in 10 shopping malls, divided into upper-end and lower-end toilets. Among the Gram-positive species, the genus Staphylococcus predominated as a typical skin bacterium. K. pneumoniae was the most frequently isolated Gram-negative species. The bacteria were isolated after cultivation on culture media and the species affiliation was determined by biotyping. Antibiotic resistance was also analyzed.

As described, there are numerous parameters that result in a higher or lower bacterial load on surfaces. The frequency of use of these restrooms plays a major role here. Is there information on how many people have visited the restrooms after cleaning and up to the time of sampling? There may be many more people out at the weekend than during the week. And, how often were the restrooms be cleaned within a day?

Response: Thank you so much for your valuable comments. Sorry for not including this crucial information. We now added some more detail about the sample collection. Restrooms are categorized as UR when situated in malls with high-end brand shops and restaurants, and as LR when located in malls with local groceries and relatively cheaper restaurants. In URs, dedicated cleaning personnel are present to clean the restroom after each use, whereas in LRs, cleaning is typically less frequent, often occur-ring at the end of the day or the beginning of the next morning. The samples that were reported were collected during the weekend between 11 am and 1 pm, which corresponds to a moderately busy period, typically considered a medium rush hour.

Is the identified antibiotic resistance due to a genetic background, i.e. do the isolates carry these genes or is it a phenotypic resistance, which is very common in P. aeruginosa. A few more details should be added and specified here.

Response: Thanks for your valuable suggestion. Unfortunately, the BD Phoenix™ automated microbiology system primarily determines phenotypic resistance patterns for antibiotic susceptibility and resistance. We now added this in our revised manuscript. However, in future we will definitely consider to determine the resistance due to genetic background too.

The frequent occurrence and dominance of Klebsiella pneumoniae is interesting. Is there an explanation for this or could this frequent occurrence also be shown in other studies?

Response: Although its not quite clear why there is frequent occurrence and dominance of Klebsiella pneumoniae in our samples in Doha, Qatar. We now added some explanation in our revised manuscript.

Table 2: requires some explanation: what does "no matching" mean for S. kloosi? What do “+++” and “+” mean and which classifications were chosen?

Response: We now added the explanation of these in our revised manuscript. The identification of Staphylococcus kloosii with a MALDI TOF score of "no matching" typically means that the MALDI-TOF system could not find a close match or similarity between the spectrum obtained from the sample and its database entries for Staphylococcus kloosii or any other known microorganism. This is not a confirmed identification.

Table 2: Change MULDI to MALDI

Response: Sorry for the typo. Changed to the revised manuscript.

The score values are relatively low in order to be able to indicate a clear species affiliation. The MALDI-TOF method also determines other closest affiliations. Were the first-named species always species-identical or were other relatives also indicated? The procedure for determining the exact species affiliation of environmental microorganisms should be considered carefully.

Response: The first-named species were always species-identical. However, we agree with the reviewer that in future, we should consider identifying the bacterial strains using 16S rRNA in parallel with MALDI-TOF system. The bacterial colonies initially identified through MALDI-TOF were subsequently subjected to confirmation testing using the BD Phoenix™ Automated Microbiology System. This system provides confirmatory identification of the tested species through a comparison of matching results from both systems.

In Tables 5 and 6, the bacterial names in the heading can be deleted as they are mentioned in the table. The species name of the bacteria should be written in lower case (Klebsiella pneumonia, not Pneumoniae)

Response: Sorry for the typo. We now corrected in the revised manuscript.

The study shows that non-pathogenic germs have been isolated, particularly in the case of staphylococci. However, antibiotic-resistant bacteria were also found. This raises the question of a risk assessment of a possible transmission of the bacteria to other people associated with a risk of infection. Are there any figures or statistics that show how many people have become infected or contaminated in toilets and are therefore at an increased risk of transmission? This should clarify whether there is a high risk of infection and therefore an urgent need for action or whether simple hygiene measures should be taken to greatly reduce the risk of infection.

Response: Thanks for pointing out a very important issue which is researchable for our future study. In another investigation, we are examining the occurrence and drug sensitivity profiles of Aeromonas bacteria isolated from green leafy vegetables and clinical samples. Our goal is to determine whether there is a correlation between the transmission of microorganisms from raw vegetables to clinical samples, or vice versa.

In what form does antibiotic resistance of a non-pathogenic microorganism play a role?

Response: We now added few lines for this impact in our revised manuscript. However, antibiotic resistance in non-pathogenic microorganisms can still play a significant role by serving as a reservoir of resistance genes and transfer these resistant genes horizontally to pathogenic bacteria, contributing to the spread of antibiotic resistance in clinical settings. They may contribute to the contamination of soil, water, or food with antibiotic-resistant genes, posing risks to human health and agricultural practices.

The different results of the upper-end and lower-end toilets are interesting. It shows that the touchless units are more hygienic. In addition to this requirement, an appeal should also be made to visitors with regard to good hygiene, as described. Another point that could certainly be included in the discussion is the installation of antimicrobial surfaces.

Response: We now included a small paragraph on utilizing the antimicrobial surfaces in designing the public restrooms in our revised manuscript. Lately, there's been a keen focus on incorporating antimicrobial surfaces into the de-sign of public restrooms. These surfaces, whether in the form of specialized materials or coatings, are specifically crafted to deter the growth and transmission of microorganisms like bacteria, viruses, and fungi on commonly used restroom surfaces. They're engineered to provide an extra layer of defense against microbial contamination, thereby elevating hygiene standards. Modern public restrooms may integrate antimicrobial features such as coatings infused with silver or copper ions, antimicrobial paints, tiles, plastics, sealants, and fabrics, all aimed at enhancing restroom hygiene.

Minor comments:

Line 77: Citrobacter and the others count as coliforms and not the other way around.

Response: Corrected in the revised manuscript.

Line 151: the nutrient agar should be specified.

Response: Included in the revised manuscript

Lines 159/168: isn't this just a repetition?

Response: Removed the repetition in the revised manuscript.

Line 320: not all bacteria mentioned are pathogenic, some are opportunistic pathogen.

Response: Corrected in the revised manuscript.

Reviewer 2 Report

Comments and Suggestions for Authors

The work addresses the well-known problem of toilet cleanliness and the potential microbiological hazard. For this reason, it does not contribute much to science other than the fact that several microorganisms are isolated and that they are antibiotic resistant. I don't know where the research was done, whether in Africa or Asia or anywhere else, there is no information about it. There is no information about the health status of the users, and the criteria for dividing toilets into lower or upper are unclear.As might be expected, the toilets identified as lower had a worse microbiological status, but it is difficult to determine whether this is due to a lack of hygiene among the people using them, a lack of or less use of disinfectants, or because more people use them. One might as well assume the false conclusion that the VIP toilet will be cleaner than the economy class passengers. I suggest to reject the manuscript.

Author Response

The work addresses the well-known problem of toilet cleanliness and the potential microbiological hazard. For this reason, it does not contribute much to science other than the fact that several microorganisms are isolated and that they are antibiotic resistant. I don't know where the research was done, whether in Africa or Asia or anywhere else, there is no information about it. There is no information about the health status of the users, and the criteria for dividing toilets into lower or upper are unclear. As might be expected, the toilets identified as lower had a worse microbiological status, but it is difficult to determine whether this is due to a lack of hygiene among the people using them, a lack of or less use of disinfectants, or because more people use them. One might as well assume the false conclusion that the VIP toilet will be cleaner than the economy class passengers. I suggest to reject the manuscript.

Response: Sorry for the unintentional mistake in not mentioning the location of the research done. The study was done in Doha, Qatar and now mentioned in the revised manuscript.

The criteria for dividing toilets into lower or upper class is now included in the revised manuscript.

Restrooms are categorized as UR when situated in malls with high-end brand shops and restaurants, and as LR when located in malls with local groceries and relatively cheaper restaurants. In URs are mostly equipped with touch-less units, dedicated cleaning personnel are present to clean the restroom after each use, whereas in LRs, cleaning is typically less frequent, often occurring at the end of the day or the beginning of the next morning. The samples that were reported were collected during the weekend between 11 am and 1 pm, which corresponds to a moderately busy period, typically considered a medium rush hour.

Apologies for any confusion regarding the objective of our study. One significant discovery from our research is that, despite lower bacterial density in URs, we identified one of the highest multi-drug resistant bacteria from HRs, even though HRs were disinfected after each use. Our intention was not to imply that HRs are inherently more hygienic than LRs or to question the general population's personal hygiene knowledge. Rather, this finding prompts further investigation into how MDR bacteria develop resistance to chemical disinfectants. Chemical disinfectants alone may not be adequate for ensuring toilet cleanliness. Incorporating knowledge of personal hygiene and antimicrobial surfaces may help mitigate the spread of infectious, disinfectant-resistant MDR bacteria. We believe this discovery will serve as a basis for studying the relationship between MDR acquisition and disinfectant resistance.

Reviewer 3 Report

Comments and Suggestions for Authors

Dear Authors,

Your article entitled "Surveillance of Bacterial Load and Multi-drug Resistant Bacteria on Surfaces of Public Restrooms" has been reviewed.

This work deserves attention since it highlights an important topic related to the identification of different pathogens and its antimicrobial resistance on surfaces of public restrooms, taking two different groups of restrooms, those of Upper-Income and those of Lower-income. This study is very important from a public health point of view.

The manuscript is well written in English, in a good design. 

Kindly fine below a list of Mino and Major Comments:

Minor Comments:

01- In the Whole manuscript, Authors are invited to put all the name bacterial "genera" or "Species" in italic. Example: Pseudomonas aeruginosa.

02- In the Whole manuscript, Authors are invited to put the full name of the bacterial species "when they used it for the first time", followed by its abbreviation between parenthesis. Example: Pseudomonas aeruginosa (P. aeruginosa)

03- Concerning the Keywords, Authors used several keywords (9), they are invited to remove 3 or 4 of these words.

04- In the Introduction section, Line 61, Authors are invited to replace "Gram-negative" by "Gram-positive".

05- In the Introduction section, Lines 89-94, when authors talked about the causative agent of each disease, they are invited to be more persuasive in term of the references used, and the name of agents causing the disease. For example:

Rotavirus (Gastroenteritis in Children). Reference: Prevalence, risk factors, and clinical characteristics of rotavirus and adenovirus among Lebanese hospitalized children with acute gastroenteritis

Staphylococcus and E. coli (Food born Diseases). Reference: Staphylococcus aureus and Staphylococcal Food-Borne Disease: An Ongoing Challenge in Public Health

E. coli, Klebsiella and Pseudomonas (UTI). Reference: Antimicrobial Susceptibilities and Laboratory Profiles of Escherichia coliKlebsiella pneumoniae, and Proteus mirabilis Isolates as Agents of Urinary Tract Infection in Lebanon: Paving the Way for Better Diagnostics

06- In the Introduction section, Line 99, this sentence needs a reference.

07- In the Materials and Methods section, Lines 134-135, Authors are invited to indicate the name of the city and the name of the country where these samples were collected.

08- Concerning the legends of Table 4, Authors are invited to replace "using MALDI-TOF-MS" by "using BD Phoenix".

09- In the Discussion section, When authors talked about Klebsiella pneumoniae, they are invited to talk about carbapenem resistance in this species (CRKP), and they can use this paper as reference for this idea: General Overview of Klebsiella pneumonia: Epidemiology and the Role of Siderophores in Its Pathogenicity   

10- Finally when authors talked about Antimicrobial resistance they are invited to talk about the importance of the discovery of new families or classes of antibiotics to continue our fight against MDR strains.  Reference: Towards new antibiotics classes targeting bacterial metallophores

Major Comments:

01- In the Results section, Lines 248-249, when you said "the six selected gram-positive isolates among the 18 isolates...", You are kindly invited to explain for readers why you select these 6 out of the 18 and not others or the full number of samples?

02- In the Results section, Lines 251-254, when you talked about the prevalence of each bacterial species in Gram positive group, you are kindly invited to explain how did you calculate these percentages since it is not clear in the manuscript.

03- In the Results section, Lines 259-261, when you talked about the prevalence of each bacterial species in Gram negative group, you are kindly invited to explain how did you calculate these percentages since it is not clear in the manuscript.

Best Regards,

Author Response

Your article entitled "Surveillance of Bacterial Load and Multi-drug Resistant Bacteria on Surfaces of Public Restrooms" has been reviewed.

This work deserves attention since it highlights an important topic related to the identification of different pathogens and its antimicrobial resistance on surfaces of public restrooms, taking two different groups of restrooms, those of Upper-Income and those of Lower-income. This study is very important from a public health point of view.

The manuscript is well written in English, in a good design. 

Kindly fine below a list of Mino and Major Comments:

Minor Comments:

01- In the Whole manuscript, Authors are invited to put all the name bacterial "genera" or "Species" in italic. Example: Pseudomonas aeruginosa.

Response: We have taken care of this issue in our revised manuscript.

02- In the Whole manuscript, Authors are invited to put the full name of the bacterial species "when they used it for the first time", followed by its abbreviation between parenthesis. Example: Pseudomonas aeruginosa (P. aeruginosa)

Response: We have taken care of this issue in our revised manuscript.

03- Concerning the Keywords, Authors used several keywords (9), they are invited to remove 3 or 4 of these words.

Response: We removed 3-4 keywords in our revised manuscript.

04- In the Introduction section, Line 61, Authors are invited to replace "Gram-negative" by "Gram-positive".

Response: sorry for the typo. We now corrected in our revised manuscript.

05- In the Introduction section, Lines 89-94, when authors talked about the causative agent of each disease, they are invited to be more persuasive in term of the references used, and the name of agents causing the disease. For example:

Rotavirus (Gastroenteritis in Children). Reference: Prevalence, risk factors, and clinical characteristics of rotavirus and adenovirus among Lebanese hospitalized children with acute gastroenteritis

Staphylococcus and E. coli (Food born Diseases). Reference: Staphylococcus aureus and Staphylococcal Food-Borne Disease: An Ongoing Challenge in Public Health

  1. coli, Klebsiella and Pseudomonas (UTI). Reference: Antimicrobial Susceptibilities and Laboratory Profiles of Escherichia coliKlebsiella pneumoniae, and Proteus mirabilisIsolates as Agents of Urinary Tract Infection in Lebanon: Paving the Way for Better Diagnostics

Response: We now added the suggested references in our revised manuscript

06- In the Introduction section, Line 99, this sentence needs a reference.

Response: Added the reference for this sentence in our revised manuscript.

07- In the Materials and Methods section, Lines 134-135, Authors are invited to indicate the name of the city and the name of the country where these samples were collected.

Response: Sorry for this mistake. We now indicated the name of the city and the name of the country where these samples were collected in our revised manuscript.

08- Concerning the legends of Table 4, Authors are invited to replace "using MALDI-TOF-MS" by "using BD Phoenix".

Response: Replaced in our revised manuscript.

09- In the Discussion section, When authors talked about Klebsiella pneumoniae, they are invited to talk about carbapenem resistance in this species (CRKP), and they can use this paper as reference for this idea: General Overview of Klebsiella pneumonia: Epidemiology and the Role of Siderophores in Its Pathogenicity   

Response: We now added the discussion of carbapenem resistance in Klebsiella pneumoniae in our revised manuscript.

10- Finally when authors talked about Antimicrobial resistance they are invited to talk about the importance of the discovery of new families or classes of antibiotics to continue our fight against MDR strains.  Reference: Towards new antibiotics classes targeting bacterial metallophores.

Response: Added to our revised manuscript.

Major Comments:

01- In the Results section, Lines 248-249, when you said "the six selected gram-positive isolates among the 18 isolates...", You are kindly invited to explain for readers why you select these 6 out of the 18 and not others or the full number of samples?

Response: Based on the morphology and biochemical tests we selected completely 6 different colonies for further automated identification and antibiotic sensitivity patterns.  

02- In the Results section, Lines 251-254, when you talked about the prevalence of each bacterial species in Gram positive group, you are kindly invited to explain how did you calculate these percentages since it is not clear in the manuscript.

Response: The percentage was calculated based on the similar colonies based on morphology and biochemical tests out of total number of colonies isolated.

03- In the Results section, Lines 259-261, when you talked about the prevalence of each bacterial species in Gram negative group, you are kindly invited to explain how did you calculate these percentages since it is not clear in the manuscript.

Response: The percentage was calculated based on the similar colonies based on morphology and biochemical tests out of total number of colonies isolated.

Round 2

Reviewer 1 Report

Comments and Suggestions for Authors

I agree with the revisions, no further comments just the two remarks:

line 31 Keywords: there are two semicolons in a row

Line 478: change K. pneumoniae in italic letters

Author Response

Comments and Suggestions for Authors

I agree with the revisions, no further comments just the two remarks:

line 31 Keywords: there are two semicolons in a row

Response: We corrected this in our revised manuscript.

Line 478: change K. pneumoniae in italic letters

Response: We corrected this in our revised manuscript.

Reviewer 2 Report

Comments and Suggestions for Authors

The corrected version is much better and could be published.

Author Response

Comments and Suggestions for Authors

The corrected version is much better and could be published.

Response: Thank you so much for your positive comments about our revised version.

Reviewer 3 Report

Comments and Suggestions for Authors

Dear Authors,

The revised manuscript was reviewed,

The article is better in the present form, 

Thank you for the modifications you made,

Best Regards,

Author Response

The revised manuscript was reviewed,

The article is better in the present form,

Thank you for the modifications you made,

Response: Thank you so much for accepting our revised version.